# ON THE CONSISTENCY OF SPHERICAL Z LOSS

**Abhishek Sharma**
kein.iitian@gmail.com

## ABSTRACT

Extremely large and sparse output space in a deep net classifier induces two major challenges of high computational complexity and class ambiguity. Class ambiguity is usually tackled by optimizing top-k error instead of zero one loss. To deal with computational complexity, recent work of Vincent et al. (2015) and de Brébisson & Vincent (2016) introduced a family of spherical loss that comes with a weight update algorithm that is independent of output space size. In this family, Z loss is of particular interest since it outperforms other spherical losses and log-softmax on top-k scores. However, there exists no theoretical result on the top-k calibration of Z loss or any concrete connection between top-k scores and hyper-parameters of Z loss. This paper provides insights on the relationship between the two and answers how and why hyper-parameters of Z loss are essential to optimize top-k scores.

## 1 INTRODUCTION

Classification problems with extremely large dimensional outputs are very common in neural language modelling tasks and becoming important in computer vision domain with the availability of large scale datasets. The most common loss optimized for training a multi-class neural networks, log-softmax, scales linearly with the number of output classes. Taking into account this bottleneck, Vincent et al. (2015) proposed an efficient algorithm for training neural networks that is independent of number of classes. However, it only works for loss functions, belonging to spherical family, that only require access to the non-zero entries in the output and the squared norm of the predicted output vector. For a detailed description of the algorithm, we refer the reader to Vincent et al. (2015). Building on this work, de Brébisson & Vincent (2016; 2017) proposed a family of spherical loss functions that satisfy the above requirements. Within this family, Z loss is of particular interest as it outperforms log softmax on large output classes benchmarks. If $\mu$ and $\sigma$ are the mean and standard deviation of multi class output vector $\boldsymbol{o}$ and c is the index of output class, de Brébisson & Vincent (2017) define Z loss as:

$$L(\boldsymbol{o}, c) = \frac{1}{a} \log \left[ 1 + \exp \left( a \left( b - \frac{o_c - \mu}{\sigma} \right) \right) \right] \qquad = \frac{1}{a} \log \left[ 1 + \exp \left( a(b - z_c) \right) \right] \quad (1)$$

where a and b are two hyper-parameters controlling the scaling of $z_c$ and $z_c = \frac{o_c - \mu}{\sigma}$. This standardization makes the Z loss invariant to both shifting and scaling of the outputs $o$ whereas log-softmax is only invariant to shifting. Empirical results on Penn Tree Bank benchmark shows that it outperforms log-softmax on various top-k scores. Besides, it delivers a significant speed up on One Billion Word Language modelling task when compared to hierarchical softmax Mikolov et al. (2013).

Although the experiments of de Brébisson & Vincent (2017) show that the choice of hyper-parameters a and b is essential to fit top-k metric empirically, no insight or an explicit relationship between the two is given. Thus, it is hard to interpret how these hyper-parameters influence the top-k scores. We show in next section that some empirical observations reported in de Brébisson & Vincent (2017) directly fall out from top-k calibration condition of Z loss.

## 2 CONSISTENCY OF Z LOSS

In statistical learning theory, a loss function is called classification calibrated if an optimal classifier with respect to the loss function yields the Bayes optimal solution ( Bartlett et al. (2003); Tewari

& Bartlett (2005); Lapin et al. (2016)). For a general multi class setting with $D$ output classes and input dimensionality $n$, one learns a function $f : \mathbb{R}^n \Rightarrow \mathbb{R}^D$ and prediction at test time is done via $\text{argmax}_{i=1..D} f_i(x)$. Lapin et al. (2016) defines a loss function to be top-k calibrated if the k largest scoring components of f corresponds to the k largest conditional class probabilities. If a loss is not top-k calibrated, it means that even in the limit of infinite data, optimizing the loss will not yield a Bayes optimal top-k error. We refer the reader to Lapin et al. (2016) for more details. We use this definition to chart a relationship between the top-k score and hyper-parameter values of Z loss.

In the following, our main goal is to check under what conditions the ordering of classifier scores is same as that of conditional class probabilities. To this end, we minimize the expected Z loss point-wise and obtain the conditions for critical points since Z loss is differentiable.

Using the tower property, we decompose the expected loss as:

$$\mathbb{E}[L(\boldsymbol{o}, \boldsymbol{f}(\boldsymbol{X}))] = \mathbb{E}[\mathbb{E}[L(\boldsymbol{o}, \boldsymbol{f}(\boldsymbol{X}))|\boldsymbol{X}]] \tag{2}$$

To compute the condition of the Bayes optimal classifier, we minimize the expected loss point-wise for each x given by:

$$\mathbb{E}[L(\boldsymbol{o}, f(X))|\boldsymbol{X} = x] = \sum_{c=1}^{D} p_x(c) L(\boldsymbol{o}, c) \tag{3}$$

Since the loss is differentiable, partial derivative of the loss w.r.t individual output activation $o_c$ is given by:

$$\frac{\partial \mathbb{E}[L]}{\partial o_c} = p_x(c) \frac{\partial L(\boldsymbol{o}, c)}{\partial o_c} + \sum_{r=1, r \neq c}^{D} p_x(r) \frac{\partial L(\boldsymbol{o}, c)}{\partial o_{r_{c \neq r}}} \tag{4}$$

Due to space constraint, we skip the derivations of first order partial derivatives of loss functions w.r.t output vector and write them directly here. They can also be verified in de Brébisson & Vincent (2017)

$$\frac{\partial L(\boldsymbol{o}, c)}{\partial o_c} = \frac{-1}{1 + \exp(a(z_c - b))} * \frac{D - 1 - z_c^2}{D\sigma} \tag{5}$$

$$\frac{\partial L(\boldsymbol{o}, c)}{\partial o_{r_{r \neq c}}} = \frac{1}{1 + \exp(a(z_r - b))} * \frac{1 + z_c z_r}{D\sigma} \tag{6}$$

Plugging the partial derivatives from Eq. 5 and Eq. 6 to Eq. 4. we get:

$$\frac{\partial \mathbb{E}[L]}{\partial o_c} = \frac{-p_x(c)}{1 + \exp(a(z_c - b))} * \frac{D - 1 - z_c^2}{D\sigma} + \sum_{r=1, r \neq c}^{D} \frac{p_x(r)}{1 + \exp(a(z_r - b))} * \frac{(1 + z_c z_r)}{D\sigma} \tag{7}$$

At any critical point, first order partial derivatives must be zero (Left side of Eq(7)),that gives the following:

$$\frac{p_x(c) * (D - 1 - z_c^2)}{1 + \exp(a(z_c - b))} = \sum_{r=1, r \neq c}^{D} \frac{p_x(r) * (1 + z_c z_r)}{1 + \exp(a(z_r - b))} \tag{8}$$

To check the the calibration condition, we impose the following ordering on the output vector:$o_{k_D} > o_{k_{D-1}} > \ldots o_{k_2} > o_{k_1}$. Given this ordering, we pick two index $k_1$ and $k_2$ and set the partial derivatives w.r.t $o_{k_1}$ and $o_{k_2}$ to zero in Eq 7:

$$\frac{p_x(k_1) * (D - 1 - z_{k_1}^2)}{1 + \exp(a(z_{k_1} - b))} = p_x(k_2) * \frac{(1 + z_{k_2} z_{k_1})}{1 + \exp(a(z_{k_2} - b))} + \sum_{k=3}^{D} \frac{p_x(k) * (1 + z_{k_1} z_k)}{1 + \exp(a(z_k - b))} \tag{9}$$

$$\frac{p_x(k_2) * (D - 1 - z_{k_2}^2)}{1 + \exp(a(z_{k_2} - b))} = p_x(k_1) * \frac{(1 + z_{k_2} z_{k_1})}{1 + \exp(a(z_{k_1} - b))} + \sum_{k=3}^{D} \frac{p_x(k) * (1 + z_{k_2} z_k)}{1 + \exp(a(z_k - b))} \tag{10}$$

Subtracting Eq. 9 from Eq. 10 and grouping $p_x(k_1)$ and $p_x(k_2)$ terms separately,

$$\frac{p_x(k_2) * (D - z_{k_2}^2 + z_{k_2} z_{k_1})}{1 + \exp(a(z_{k_2} - b))} = \frac{p_x(k_1) * (D + z_{k_1} z_{k_2} - z_{k_1}^2)}{1 + \exp(a(z_{k_1} - b))} + \sum_{k=3}^{D} \frac{p_x(k) * (1 + z_k(z_{k_2} - z_{k_1}))}{1 + \exp(a(z_k - b))} \tag{11}$$

$$\frac{p_x(k_2) * (D - z_{k_2}(z_{k_2} - z_{k_1}))}{1 + \exp(a(z_{k_2} - b))} = \frac{p_x(k_1) * (D + z_{k_1}(z_{k_2} - z_{k_1}))}{1 + \exp(a(z_{k_1} - b))} + \sum_{k=3}^{D} \frac{p_x(k) * (1 + z_k(z_{k_2} - z_{k_1}))}{1 + \exp(a(z_k - b))} \tag{12}$$

$$D * \left( \frac{p_x(k_2)}{1 + \exp(a(z_{k_2} - b))} - \frac{p_x(k_1)}{1 + \exp(a(z_{k_1} - b))} \right) = \sum_{k=1}^{D} \frac{p_x(k) * (1 + z_k(z_{k_2} - z_{k_1}))}{1 + \exp(a(z_k - b))} \tag{13}$$

$$\frac{p_x(k_2)}{1 + \exp(a(z_{k_2} - b))} - \frac{p_x(k_1)}{1 + \exp(a(z_{k_1} - b))} = \frac{1}{D} * \sum_{k=1}^{D} \frac{p_x(k) * (1 + z_k(z_{k_2} - z_{k_1}))}{1 + \exp(a(z_k - b))} \tag{14}$$

Note that exponential is a strict monotonic function. For $o_{k_2} > o_{k_1}$, we have $z_{k_2} > z_{k_1}$, by definition. Thus, summation in right side of Eq. 14 is always positive for $a > 0$ and $z_k > 0$. The positive right side implies that on left side, $p_x(k_2) > p_x(k_1)$ since denominator of $p_x(k_2) > p_x(k_1)$ for $a > 0$ and $z_k > 0$. It is easy to deduce that the Eq. 14 holds for any other two conditional class probabilities.

## 2.1 DISCUSSION

Since $z_k$ are normalized and scaled activation output values, some $z_k$ values will inevitably be negative. Thus, to maintain the same ordering of conditional class probabilities, denominators in Eq. 14 play a crucial role. In particular, value of $a$, in left hand side of Eq. 14, is critical to ensure that $p_x(k_2) > p_x(k_1)$ if the summation in right hand side turns out to be negative. This also implies that we only need to fit hyper-parameter $a$ value to optimize top-k score since $(z_{k_2} - b) > (z_{k_1} - b)$ for any constant $b$. This observation is in line with the empirical findings of de Brébisson & Vincent (2017) where they report that hyper-parameter $a$ is more important than $b$ while optimizing for top-k scores.

## 3 CONCLUSION

We provide an explicit relationship that answers how and why hyper-parameters of Z loss are essential to optimize top-k scores. Our result also suggest that one only needs to tune value of hyper-parameter $a$ to better optimize the top-k score. Despite the dependency on the sign of $z_k$ in the resulting Z loss calibration condition , it is also interesting to note that it outperforms log softmax on top-k scores even though log softmax is shown to be top-k calibrated for all k by Lapin et al. (2016). Since our work relies on computing critical points without any guarantee of a global minimum, we believe proposing a convex upper bound of Z loss, while retaining its important properties such as scaling and shifting, is a promising future research direction.

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
