# OpenReview forum: "On the Consistency of Spherical Z Loss"
_ICLR.cc/2018/Workshop — Reject_

### Official Review · AnonReviewer2 · 2018-03-08
**Investigate relationship on how and why hyper-parameters of Z loss are essential to optimize top-k scores.**

**Rating:** 4
**Confidence:** 4

**Review:**

The paper is very badly written and lacks rigor at any analytical depth.
The derivations are tedious but quite immediate and easy.
Symbols are badly used.

---

### Official Review · AnonReviewer3 · 2018-03-11
**Correct, but not a very exciting or surprising result**

**Rating:** 6
**Confidence:** 4

**Review:**

This paper sheds some light on the relationship between the top-k calibration property and the hyper-parameters of the Z loss. The derivation is correct, but the result is not surprising and doesn't have any strong important consequences; the exception may be that the result suggests that only one of the two hyper-parameters of the Z loss needs to be tuned in order to optimize the top-k score.

In terms of writing, the paper is missing the word "the" many times (in addition to other grammatical errors). For example: "Within this family, Z loss is of particular interest..." should be "Within this family, the Z loss is of particular interest"; "...where a and b are two hyper-parameters controlling..." should be "...where a and b are the two hyper-parameters controlling..."; "Empirical results on Penn Tree Bank benchmark shows..." should be "Empirical results on the Penn Tree Bank benchmark shows..."

---

### Official Review · AnonReviewer1 · 2018-03-11
**Statistical consistency of Z-loss**

**Rating:** 4
**Confidence:** 4

**Review:**

This paper attempts to analyze the statistical consistency (namely, top-k consistecy) of the recently proposed spherical Z-loss, a loss function with some appealing properties that is an alternative to softmax.

Unfortunately, the presentation lacks clarity and motivation for the several derivation steps, and the theoretical analysis seems to fall short. Notation is imprecise in some parts, and I have some doubts about the overall technical correctness.

The expression in n Eq 4 is erroneous and may impact the technical correctness of the derivation steps that follow. The righthand term should be

\sum_{r \ne c} p_x(r) \frac{\nabla L(o, r)}{\nabla o_c}

instead of

\sum_{r \ne c} p_x(r) \frac{\nabla L(o, c)}{\nabla o_r}.

I don't know how this affects the correctness of Eq 7 (since I don't know if Eq 6 has the same problem), namely if the denominator inside the sum in Eq 7 should be 1 + exp(a(z_c - b)) instead of 1 + exp(a(z_r - b)).

The paper doesn't explain why it's trying to compute the "critical points" of the expected loss. Some motivation should be given, namely that the points o with zero gradient can correspond to minimizers of the loss for the given "true" distribution p_x(c). This will help the reader understand why the author is doing this.

The author then arrives at Eq 14, but unfortunately I don't think there's much we can conclude from this equation. As the author points out, the summation in the RHS is only guaranteed to be positive if a>0 (which is fine) and if z_k>0 (which will not happen for several k's, given that z is centered and standardized). What can we conclude for k's for which z_k is negative? The discussion argues that the choice of the hyperparameter "a" can fix this, but can it? It seems that there is no single "a" that can fix this for any possible k and p_x(c).

In my opinion, this is a "half-baked" work and it's hard to understand what the take-home message of this contribution should be.

Minor comments:
- use $a$, $b$, $k$, instead of a, b, k, when to refering to these variables inline.
- remove the long space before the "=" sign in Eq 1
- in Eq 3, it should be stated that p_x(c) means p(c | x)
- in Eq 4, the shorthand E[L] for E[L(o, f(X) | X=x)] has not been explained

---

### Decision · Program_Chairs · 2018-03-20
**ICLR 2018 Workshop Acceptance Decision**

**Decision:**

Reject

**Comment:**

Based on the reviews, this paper has not been accepted for presentation at the ICLR workshop. However, the conversation and updates can continue to appear here on OpenReview.